# Towards Robust Interpretability
# with Self-Explaining Neural Networks

**David Alvarez-Melis**
CSAIL, MIT
dalvmel@mit.edu

**Tommi S. Jaakkola**
CSAIL, MIT
tommi@csail.mit.edu

## Abstract

Most recent work on interpretability of complex machine learning models has focused on estimating *a posteriori* explanations for previously trained models around specific predictions. *Self-explaining* models where interpretability plays a key role already during learning have received much less attention. We propose three desiderata for explanations in general – explicitness, faithfulness, and stability – and show that existing methods do not satisfy them. In response, we design self-explaining models in stages, progressively generalizing linear classifiers to complex yet architecturally explicit models. Faithfulness and stability are enforced via regularization specifically tailored to such models. Experimental results across various benchmark datasets show that our framework offers a promising direction for reconciling model complexity and interpretability.

## 1 Introduction

Interpretability or lack thereof can limit the adoption of machine learning methods in decision-critical —e.g., medical or legal— domains. Ensuring interpretability would also contribute to other pertinent criteria such as fairness, privacy, or causality [5]. Our focus in this paper is on complex *self-explaining* models where interpretability is built-in architecturally and enforced through regularization. Such models should satisfy three desiderata for interpretability: explicitness, faithfulness, and stability where, for example, stability ensures that similar inputs yield similar explanations. Most post-hoc interpretability frameworks are not stable in this sense as shown in detail in Section 5.4.

High modeling capacity is often necessary for competitive performance. For this reason, recent work on interpretability has focused on producing *a posteriori* explanations for performance-driven deep learning approaches. The interpretations are derived locally, around each example, on the basis of limited access to the inner workings of the model such as gradients or reverse propagation [4, 18], or through oracle queries to estimate simpler models that capture the local input-output behavior [16, 2, 14]. Known challenges include the definition of locality (e.g., for structured data [2]), identifiability [12] and computational cost (with some of these methods requiring a full-fledged optimization subroutine [24]). However, point-wise interpretations generally do not compare explanations obtained for nearby inputs, leading to unstable and often contradicting explanations [1].

A posteriori explanations may be the only option for already-trained models. Otherwise, we would ideally design the models from the start to provide human-interpretable explanations of their predictions. In this work, we build highly complex interpretable models bottom up, maintaining the desirable characteristics of simple linear models in terms of features and coefficients, without limiting performance. For example, to ensure stability (and, therefore, interpretability), coefficients in our model vary slowly around each input, keeping it effectively a linear model, albeit locally. In other words, our model operates as a simple interpretable model *locally* (allowing for point-wise interpretation) *but not globally* (which would entail sacrificing capacity). We achieve this with a regularization scheme that ensures our model not only *looks* like a linear model, but (locally) *behaves* like one.

Our main contributions in this work are:

- A rich class of interpretable models where the explanations are intrinsic to the model
- Three desiderata for explanations together with an optimization procedure that enforces them
- Quantitative metrics to empirically evaluate whether models adhere to these three principles, and showing the advantage of the proposed self-explaining models under these metrics

## 2  Interpretability: linear and beyond

To motivate our approach, we start with a simple linear regression model and successively generalize it towards the class of self-explaining models. For input features $x_1, \ldots, x_n \in \mathbb{R}$, and associated parameters $\theta_0, \ldots, \theta_n \in \mathbb{R}$ the linear regression model is given by $f(x) = \sum_i^n \theta_i x_i + \theta_0$. This model is arguably interpretable for three specific reasons: i) input features ($x_i$'s) are clearly anchored with the available observations, e.g., arising from empirical measurements; ii) each parameter $\theta_i$ provides a quantitative positive/negative contribution of the corresponding feature $x_i$ to the predicted value; and iii) the aggregation of feature specific terms $\theta_i x_i$ is additive without conflating feature-by-feature interpretation of impact. We progressively generalize the model in the following subsections and discuss how this mechanism of interpretation is preserved.

### 2.1  Generalized coefficients

We can substantially enrich the linear model while keeping its overall structure if we permit the coefficients themselves to depend on the input $x$. Specifically, we define (offset function omitted) $f(x) = \theta(x)^\top x$, and choose $\theta$ from a complex model class $\Theta$, realized for example via deep neural networks. Without further constraints, the model is nearly as powerful as—and surely no more interpretable than—any deep neural network. However, in order to maintain interpretability, at least locally, we must ensure that for close inputs $x$ and $x'$ in $\mathbb{R}^n$, $\theta(x)$ and $\theta(x')$ should not differ significantly. More precisely, we can, for example, regularize the model in such a manner that $\nabla_x f(x) \approx \theta(x_0)$ for all $x$ in a neighborhood of $x_0$. In other words, the model acts locally, around each $x_0$, as a linear model with a vector of stable coefficients $\theta(x_0)$. The individual values $\theta(x_0)_i$ act and are interpretable as coefficients of a linear model with respect to the final prediction, but adapt dynamically to the input, albeit varying slower than $x$. We will discuss specific regularizers so as to keep this interpretation in Section 3.

### 2.2  Beyond raw features – feature basis

Typical interpretable models tend to consider each variable (one feature or one pixel) as the fundamental unit which explanations consist of. However, pixels are rarely the basic units used in human image understanding; instead, we would rely on strokes and other higher order features. We refer to these more general features as *interpretable basis concepts* and use them in place of raw inputs in our models. Formally, we consider functions $h(x) : \mathcal{X} \to \mathcal{Z} \subset \mathbb{R}^k$, where $\mathcal{Z}$ is some space of interpretable atoms. Naturally, $k$ should be small so as to keep the explanations easily digestible. Alternatives for $h(\cdot)$ include: (i) subset aggregates of the input (e.g., with $h(x) = Ax$ for a boolean mask matrix $A$), (ii) predefined, pre-grounded feature extractors designed with expert knowledge (e.g., filters for image processing), (iii) prototype based concepts, e.g. $h(x)_i = \|x - \xi_i\|$ for some $\xi_i \in \mathcal{X}$ [12], or (iv) learnt representations with specific constraints to ensure grounding [19]. Naturally, we can let $h(x) = x$ to recover raw-input explanations if desired. The generalized model is now:

$$f(x) = \theta(x)^\top h(x) = \sum_{i=1}^k \theta(x)_i h(x)_i \tag{1}$$

Since each $h(x)_i$ remains a scalar, it can still be interpreted as the degree to which a particular feature is present. In turn, with constraints similar to those discussed above $\theta(x)_i$ remains interpretable as a local coefficient. Note that the notion of locality must now take into account how the concepts rather than inputs vary since the model is interpreted as being linear in the concepts rather than $x$.

### 2.3  Further generalization

The final generalization we propose considers how the elements $\theta(x)_i h(x)_i$ are aggregated. We can achieve a more flexible class of functions by replacing the sum in (1) by a more general aggregation

function $g(z_1, \ldots, z_k)$, where $z_i := \theta(x)_i h(x)_i$. Naturally, in order for this function to preserve the desired interpretation of $\theta(x)$ in relation to $h(x)$, it should: i) be permutation invariant, so as to eliminate higher order uninterpretable effects caused by the relative position of the arguments, (ii) isolate the effect of individual $h(x)_i$'s in the output (e.g., avoiding multiplicative interactions between them), and (iii) preserve the sign and relative magnitude of the impact of the relevance values $\theta(x)_i$. We formalize these intuitive desiderata in the next section.

Note that we can naturally extend the framework presented in this section to multivariate functions with range in $\mathcal{Y} \subset \mathbb{R}^m$, by considering $\theta_i : \mathcal{X} \to \mathbb{R}^m$, so that $\theta_i(x) \in \mathbb{R}^m$ is a vector corresponding to the relevance of concept $i$ with respect to each of the $m$ output dimensions. For classification, however, we are mainly interested in the explanation for the predicted class, i.e., $\theta_{\hat{y}}(x)$ for $\hat{y} = \operatorname{argmax}_y p(y|x)$.

## 3 Self-explaining models

We now formalize the class of models obtained through subsequent generalization of the simple linear predictor in the previous section. We begin by discussing the properties we wish to impose on $\theta$ in order for it to act as coefficients of a linear model on the basis concepts $h(x)$. The intuitive notion of robustness discussed in Section 2.2 suggests using a condition bounding $\|\theta(x) - \theta(y)\|$ with $L\|h(x) - h(y)\|$ for some constant $L$. Note that this resembles, but is not exactly equivalent to, Lipschitz continuity, since it bounds $\theta$'s variation with respect to a different—and indirect—measure of change, provided by the geometry induced implicitly by $h$ on $\mathcal{X}$. Specifically,

**Definition 3.1.** *We say that a function $f : \mathcal{X} \subseteq \mathbb{R}^n \to \mathbb{R}^m$ is **difference-bounded** by $h : \mathcal{X} \subseteq \mathbb{R}^n \to \mathbb{R}^k$ if there exists $L \in \mathbb{R}$ such that $\|f(x) - f(y)\| \le L\|h(x) - h(y)\|$ for every $x, y \in \mathcal{X}$.*

Imposing such a global condition might be undesirable in practice. The data arising in applications often lies on low dimensional manifolds of irregular shape, so a uniform bound might be too restrictive. Furthermore, we specifically want $\theta$ to be consistent *for neighboring inputs*. Thus, we seek instead a *local* notion of stability. Analogous to the local Lipschitz condition, we propose a pointwise, neighborhood-based version of Definition 3.1:

**Definition 3.2.** $f : \mathcal{X} \subseteq \mathbb{R}^n \to \mathbb{R}^m$ *is **locally difference-bounded** by $h : \mathcal{X} \subseteq \mathbb{R}^n \to \mathbb{R}^k$ if for every $x_0$ there exist $\delta > 0$ and $L \in \mathbb{R}$ such that $\|x - x_0\| < \delta$ implies $\|f(x) - f(x_0)\| \le L\|h(x) - h(x_0)\|$.*

Note that, in contrast to Definition 3.1, this second notion of stability allows $L$ (and $\delta$) to depend on $x_0$, that is, the "Lipschitz" constant can vary throughout the space. With this, we are ready to define the class of functions which form the basis of our approach.

**Definition 3.3.** *Let $x \in \mathcal{X} \subseteq \mathbb{R}^n$ and $\mathcal{Y} \subseteq \mathbb{R}^m$ be the input and output spaces. We say that $f : \mathcal{X} \to \mathcal{Y}$ is a **self-explaining prediction model** if it has the form*

$$f(x) = g\big(\theta(x)_1 h(x)_1, \ldots, \theta(x)_k h(x)_k\big) \tag{2}$$

*where:*

*P1) $g$ is monotone and completely additively separable*
*P2) For every $z_i := \theta(x)_i h(x)_i$, $g$ satisfies $\frac{\partial g}{\partial z_i} \ge 0$*
*P3) $\theta$ is locally difference-bounded by $h$*
*P4) $h(x)$ is an interpretable representation of $x$*
*P5) $k$ is small.*

*In that case, for a given input $x$, we define the explanation of $f(x)$ to be the set $\mathcal{E}_f(x) \triangleq \{(h(x)_i, \theta(x)_i)\}_{i=1}^k$ of basis concepts and their influence scores.*

Besides the linear predictors that provided a starting point in Section 2, well-known families such as generalized linear models and nearest-neighbor classifiers are contained in this class of functions. However, the true power of the models described in Definition 3.3 comes when $\theta(\cdot)$ (and potentially $h(\cdot)$) are realized by architectures with large modeling capacity, such as deep neural networks. When $\theta(\cdot)$ is realized with a neural network, we refer to $f$ as a *self-explaining neural network* (SENN). If $g$ depends on its arguments in a continuous way, $f$ can be trained end-to-end with back-propagation. Since our aim is maintaining model richness even in the case where the concepts are chosen to be raw inputs (i.e., $h$ is the identity), we rely predominantly on $\theta$ for modeling capacity, realizing it with larger, higher-capacity architectures.

It remains to discuss how the properties (P1)-(P5) in Definition 3.3 are to be enforced. The first two depend entirely on the choice of aggregating function $g$. Besides trivial addition, other options include affine functions $g(z_1, \ldots, z_k) = \sum_i A_i z_i$ where the $A_i$ are constrained to be positive. On the other hand, the last two conditions in Definition 3.3 are application-dependent: what and how many basis concepts are adequate should be informed by the problem and goal at hand.

The only condition in Definition 3.3 that warrants further discussion is (P3): the stability of $\theta$ with respect to $h$. For this, let us consider what $f$ would look like if the $\theta_i$'s were indeed (constant) parameters. Looking at $f$ as a function of $h(x)$, i.e., $f(x) = g(h(x))$, let $z = h(x)$. Using the chain rule we get $\nabla_x f = \nabla_z f \cdot J_x^h$, where $J_x^h$ denotes the Jacobian of $h$ (with respect to $x$). At a given point $x_0$, we want $\theta(x_0)$ to behave as the derivative of $f$ with respect to the concept vector $h(x)$ around $x_0$, i.e., we seek $\theta(x_0) \approx \nabla_z f$. Since this is hard to enforce directly, we can instead plug this *ansatz* in $\nabla_x f = \nabla_z f \cdot J_x^h$ to obtain a proxy condition:

$$\mathcal{L}_\theta(f(x)) \triangleq \|\nabla_x f(x) - \theta(x)^\top J_x^h(x)\| \approx 0 \tag{3}$$

All three terms in $\mathcal{L}_\theta(f)$ can be computed, and when using differentiable architectures $h(\cdot)$ and $\theta(\cdot)$, we obtain gradients with respect to (3) through automatic differentiation and thus use it as a regularization term in the optimization objective. With this, we obtain a gradient-regularized objective of the form $\mathcal{L}_y(f(x), y) + \lambda \mathcal{L}_\theta(f(x))$, where the first term is a classification loss and $\lambda$ a parameter that trades off performance against stability —and therefore, interpretability— of $\theta(x)$.

## 4 Learning interpretable basis concepts

Raw input features are the natural basis for interpretability when the input is low-dimensional and individual features are meaningful. For high-dimensional inputs, raw features (such as individual pixels in images) tend to be hard to analyze coherently and often lead to unstable explanations that are sensitive to noise or imperceptible artifacts in the data [1], and not robust to simple transformations such as constant shifts [9]. The results in the next section confirm this phenomenon, where we observe that the lack of robustness of methods that rely on raw inputs is amplified for high-dimensional inputs. To avoid some of these shortcomings, we can instead operate on higher level features. In the context of images, we might be interested in the effect of textures or shapes—rather than single pixels—on predictions. For example, in medical image processing higher-level visual aspects such as tissue ruggedness, irregularity or elongation are strong predictors of cancerous tumors, and are among the first aspects that doctors look for when diagnosing, so they are natural "units" of explanation.

Ideally, these basis concepts would be informed by expert knowledge, such as the doctor-provided features mentioned above. However, in cases where such prior knowledge is not available, the basis concepts could be learnt instead. Interpretable concept learning is a challenging task in its own right [8], and as other aspects of interpretability, remains ill-defined. We posit that a reasonable minimal set of desiderata for interpretable concepts is:

i) **Fidelity**: the representation of $x$ in terms of concepts should preserve relevant information,

ii) **Diversity**: inputs should be representable with few non-overlapping concepts, and

iii) **Grounding**: concepts should have an immediate human-understandable interpretation.

Here, we enforce these conditions upon the concepts learnt by SENN by: (i) training $h$ as an autoencoder, (ii) enforcing diversity through sparsity and (iii) providing interpretation on the concepts by prototyping (e.g., by providing a small set of training examples that maximally activate each concept, as described below). Learning of $h$ is done end-to-end in conjunction with the rest of the model. If we denote by $h_{dec}(\,\cdot\,) : \mathbb{R}^k \to \mathbb{R}^n$ the decoder associated with $h$, and $\hat{x} := h_{dec}(h(x))$ the reconstruction of $x$, we use an additional penalty $\mathcal{L}_h(x, \hat{x})$ on the objective, yielding the loss:

$$\mathcal{L}_y(f(x), y) + \lambda \mathcal{L}_\theta(f(x)) + \xi \mathcal{L}_h(x, \hat{x}) \tag{4}$$

Achieving (iii), i.e., the grounding of $h(x)$, is more subjective. A simple approach consists of representing each concept by the elements in a sample of data that maximize their value, that is, we can represent concept $i$ through the set $X^i = \operatorname{argmax}_{\hat{X} \subseteq X, |\hat{X}|=l} \sum_{x \in \hat{X}} h(x)_i$ where $l$ is small. Similarly, one could construct (by optimizing $h$) synthetic inputs that maximally activate each concept (and do not activate others), i.e., $\operatorname{argmax}_{x \in \mathcal{X}} h_i(x) - \sum_{j \neq i} h_j(x)$. Alternatively, when available, one might want to represent concepts via their learnt weights—e.g., by looking at the filters associated with each concept in a CNN-based $h(\,\cdot\,)$. In our experiments, we use the first of these approaches (i.e., using maximally activated prototypes), leaving exploration of the other two for future work.

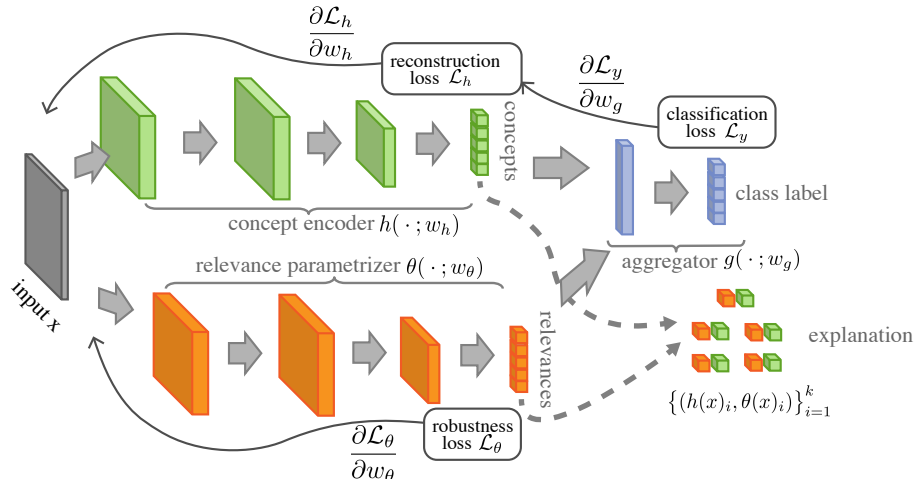

Figure 1: A SENN consists of three components: a **concept encoder** (green) that transforms the input into a small set of interpretable basis features; an **input-dependent parametrizer** (orange) that generates relevance scores; and an **aggregation function** that combines to produce a prediction. The robustness loss on the parametrizer encourages the full model to behave locally as a linear function on $h(x)$ with parameters $\theta(x)$, yielding immediate interpretation of both concepts and relevances.

## 5 Experiments

The notion of interpretability is notorious for eluding easy quantification [5]. Here, however, the motivation in Section 2 produced a set of desiderata according to which we can validate our models. Throughout this section, we base the evaluation on four main criteria. First and foremost, for all datasets we investigate whether our models perform on par with their non-modular, non interpretable counterparts. After establishing that this is indeed the case, we focus our evaluation on the *interpretability* of our approach, in terms of three criteria:

(i) **Explicitness/Intelligibility**: *Are the explanations immediate and understandable?*

(ii) **Faithfulness**: *Are relevance scores indicative of "true" importance?*

(iii) **Stability**: *How consistent are the explanations for similar/neighboring examples?*

Below, we address these criteria one at a time, proposing qualitative assessment of (i) and quantitative metrics for evaluating (ii) and (iii).

### 5.1 Dataset and Methods

**Datasets** We carry out quantitative evaluation on three classification settings: (i) MNIST digit recognition, (ii) benchmark UCI datasets [13] and (iii) Propublica's COMPAS Recidivism Risk Score datasets.[1] In addition, we provide some qualitative results on CIFAR10 [10] in the supplement (§A.5). The COMPAS data consists of demographic features labeled with criminal *recidivism* ("relapse") risk scores produced by a private company's proprietary algorithm, currently used in the Criminal Justice System to aid in bail granting decisions. Propublica's study showing racial-biased scores sparked a flurry of interest in the COMPAS algorithm both in the media and in the fairness in machine learning community [25, 7]. Details on data pre-processing for all datasets are provided in the supplement.

**Comparison methods.** We compare our approach against various interpretability frameworks: three popular "black-box" methods; LIME [16], kernel Shapley values (SHAP, [14]) and perturbation-based occlusion sensitivity (OCCLUSION) [26]; and various gradient and saliency based methods: gradient×input (GRAD*INPUT) as proposed by Shrikumar et al. [20], saliency maps (SALIENCY) [21], Integrated Gradients (INT.GRAD) [23] and ($\epsilon$)-Layerwise Relevance Propagation (E-LRP) [4].

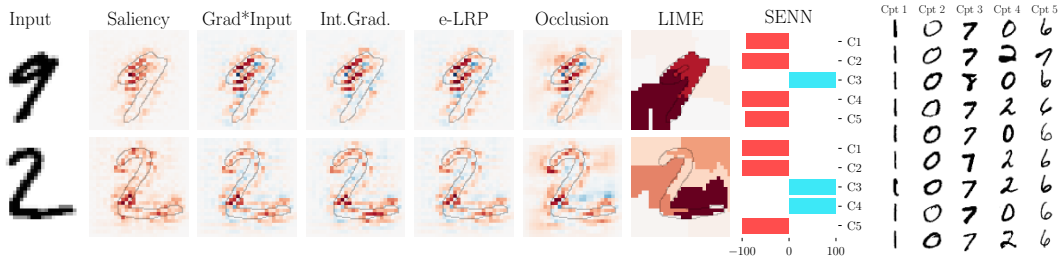

Figure 2: A comparison of traditional input-based explanations (positive values depicted in red) and SENN's concept-based ones for the predictions of an image classification model on MNIST. The explanation for SENN includes a characterization of concepts in terms of defining prototypes.

## 5.2 Explicitness/Intelligibility: How understandable are SENN's explanations?

When taking $h(x)$ to be the identity, the explanations provided by our method take the same surface level (i.e, heat maps on inputs) as those of common saliency and gradient-based methods, but differ substantially when using concepts as a unit of explanations (i.e., $h$ is learnt). In Figure 2 we contrast these approaches in the context of digit classification interpretability. To highlight the difference, we use only a handful of concepts, forcing the model encode digits into meta-types sharing higher level information. Naturally, it is necessary to describe each concept to understand what it encodes, as we do here through a grid of the most representative prototypes (as discussed in §4), shown here in Fig. 2, right. While pixel-based methods provide more granular information, SENN's explanation is (by construction) more parsimonious. For both of these digits, Concept 3 had a strong positive influence towards the prediction. Indeed, that concept seems to be associated with diagonal strokes (predominantly occurring in 7's), which both of these inputs share. However, for the second prediction there is another relevant concept, C4, which is characterized largely by stylized 2's, a concept that in contrast has negative influence towards the top row's prediction.

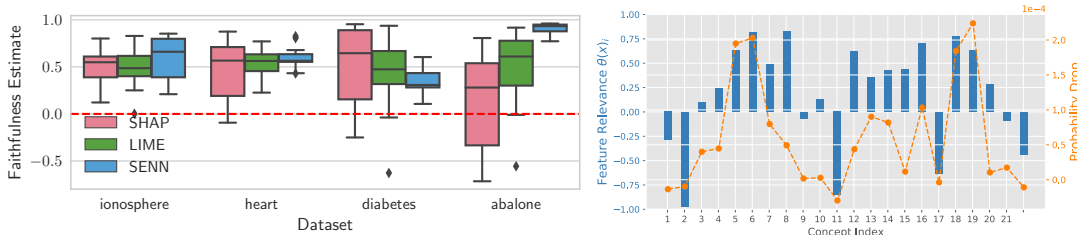

Figure 3: **Left**: Aggregated correlation between feature relevance scores and true importance, as described in Section 5.3. **Right**: Faithfulness evaluation SENN on MNIST with learnt concepts.

## 5.3 Faithfulness: Are "relevant" features truly relevant?

Assessing the correctness of estimated feature relevances requires a reference "true" influence to compare against. Since this is rarely available, a common approach to measuring the *faithfulness* of relevance scores with respect to the model they are explaining relies on a proxy notion of importance: observing the effect of removing features on the model's prediction. For example, for a probabilistic classification model, we can obscure or remove features, measure the drop in probability of the predicted class, and compare against the interpreter's own prediction of relevance [17, 3]. Here, we further compute the correlations of these probability drops and the relevance scores on various points, and show the aggregate statistics in Figure 3 (left) for LIME, SHAP and SENN (without learnt concepts) on various UCI datasets. We note that this evaluation naturally extends to the case where the concepts are learnt (Fig. 3, right). The additive structure of our model allows for *removal* of features $h(x)_i$—regardless of their form, i.e., inputs or concepts—simply by setting their coefficients $\theta_i$ to zero. Indeed, while feature removal is not always meaningful for other predictions models (i.e., one must replace pixels with black or averaged values to simulate removal in a CNN), the definition of our model allows for targeted removal of features, rendering an evaluation based on it more reliable.

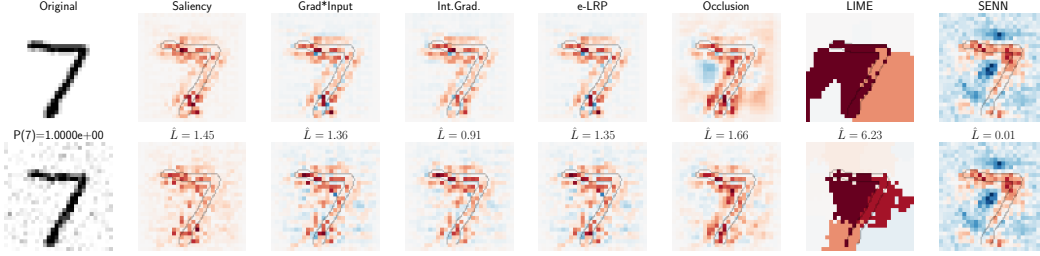

Figure 4: Explaining a CNN's prediction on an true MNIST digit (top row) and a perturbed version with added Gaussian noise. Although the model's prediction is mostly unaffected by this perturbation (change in prediction probability $\leq 10^{-4}$), the explanations for post-hoc methods vary considerably.

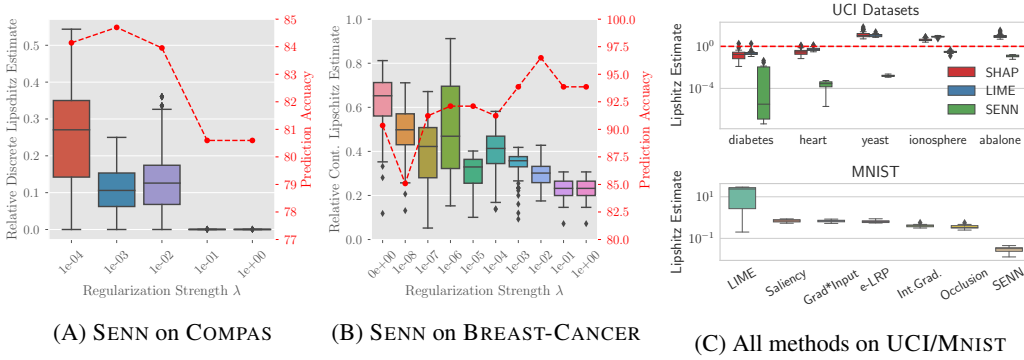

(A) SENN on COMPAS   (B) SENN on BREAST-CANCER   (C) All methods on UCI/MNIST

Figure 5: **(A/B)**: Effect of regularization on SENN's performance. **(C)**: Robustness comparison.

## 5.4   Stability: How coherent are explanations for similar inputs?

As argued throughout this work, a crucial property that interpretability methods should satisfy to generate meaningful explanations is that of robustness with respect to local perturbations of the input. Figure 4 shows that this is not the case for popular interpretability methods; even adding minimal white noise to the input introduces visible changes in the explanations. But to formally quantify this phenomenon, we appeal again to Definition 3.2 as we seek a worst-case (*adversarial*) notion of robustness. Thus, we can quantify the stability of an explanation generation model $f_{\text{expl}}(x)$, by estimating, for a given input $x$ and neighborhood size $\epsilon$:

$$\hat{L}(x_i) = \underset{x_j \in B_\epsilon(x_i)}{\text{argmax}} \frac{\|f_{\text{expl}}(x_i) - f_{\text{expl}}(x_j)\|_2}{\|h(x_i) - h(x_j)\|_2} \tag{5}$$

where for SENN we have $f_{\text{expl}}(x) := \theta(x)$, and for raw-input methods we replace $h(x)$ with $x$, turning (5) into an estimation of the Lipschitz constant (in the usual sense) of $f_{\text{expl}}$. We can directly estimate this quantity for SENN since the explanation generation is end-to-end differentiable with respect to concepts, and thus we can rely on direct automatic differentiation and back-propagation to optimize for the maximizing argument $x_j$, as often done for computing adversarial examples for neural networks [6]. Computing (5) for post-hoc explanation frameworks is, however, much more challenging, since they are not end-to-end differentiable. Thus, we need to rely on black-box optimization instead of gradient ascent. Furthermore, evaluation of $f_{\text{expl}}$ for methods like LIME and SHAP is expensive (as it involves model estimation for each query), so we need to do so with a restricted evaluation budget. In our experiments, we rely on Bayesian Optimization [22].

The *continuous* notion of local stability (5) might not be suitable for discrete inputs or settings where adversarial perturbations are overly restrictive (e.g., when the true data manifold has regions of flatness in some dimensions). In such cases, we can instead define a (weaker) sample-based notion of stability. For any $x$ in a finite sample $X = \{x_i\}_{i=1}^n$, let its $\epsilon$-neighborhood within $X$ be $\mathcal{N}_\epsilon(x) = \{x' \in X \mid \|x - x'\| \leq \epsilon\}$. Then, we consider an alternative version of (5) with $\mathcal{N}_\epsilon(x)$ in lieu of $B_\epsilon(x_i)$. Unlike the former, its computation is trivial since it involves a finite sample.

We first use this evaluation metric to validate the usefulness of the proposed gradient regularization approach for enforcing explanation robustness. The results on the COMPAS and BREAST-CANCER datasets (Fig. 5 A/B), show that there is a natural tradeoff between stability and prediction accuracy through the choice of regularization parameter $\lambda$. Somewhat surprisingly, we often observe an boost in performance brought by the gradient penalty, likely caused by the additional regularization it imposes on the prediction model. We observe a similar pattern on MNIST (Figure 8, in the Appendix). Next, we compare all methods in terms of robustness on various datasets (Fig. 5C), where we observe SENN to consistently and substantially outperform all other methods in this metric.

It is interesting to visualize the inputs and corresponding explanations that maximize criterion (5) –or its discrete counterpart, when appropriate– for different methods and datasets, since these succinctly exhibit the issue of lack of robustness that our work seeks to address. We provide many such "adversarial" examples in Appendix A.7. These examples show the drastic effect that minimal perturbations can have on most methods, particularly LIME and SHAP. The pattern is clear: most current interpretability approaches are not robust, even when the underlying model they are trying to explain is. The class of models proposed here offers a promising avenue to remedy this shortcoming.

## 6 Related Work

**Interpretability methods for neural networks**. Beyond the gradient and perturbation-based methods mentioned here [21, 26, 4, 20, 23], various other methods of similar spirit exist [15]. These methods have in common that they do not modify existing architectures, instead relying on a-posteriori computations to reverse-engineer importance values or sensitivities of inputs. Our approach differs both in what it considers the *units* of explanation—general concepts, not necessarily raw inputs—and how it uses them, intrinsically relying on the relevance scores it produces to make predictions, obviating the need for additional computation. More related to our approach is the work of Lei et al. [11] and Al-Shedivat et al. [19]. The former proposes a neural network architecture for text classification which "justifies" its predictions by selecting relevant tokens in the input text. But this *interpretable* representation is then operated on by a complex neural network, so the method is transparent as to *what* aspect of the input it uses for prediction, but not *how* it uses it. Contextual Explanation Networks [19] are also inspired by the goal of designing a class of models that learns to predict and explain jointly, but differ from our approach in their formulation (through deep graphical models) and realization of the model (through variational autoencoders). Furthermore, our approach departs from that work in that we explicitly enforce robustness with respect to the units of explanation and we formulate concepts as part of the explanation, thus requiring them to be grounded and interpretable.

**Explanations through concepts and prototypes.** Li et al. [12] propose an interpretable neural network architecture whose predictions are based on the similarity of the input to a small set of prototypes, which are learnt during training. Our approach can be understood as generalizing this approach beyond similarities to prototypes into more general interpretable concepts, while differing in how these higher-level representation of the inputs are used. More similar in spirit to our approach of explaining by means of learnable interpretable concepts is the work of Kim et al. [8]. They propose a technique for learning *concept activation vectors* representing human-friendly concepts of interest, by relying on a set of human-annotated examples characterizing these. By computing directional derivatives along these vectors, they gauge the sensitivity of predictors with respect to *semantic* changes in the direction of the concept. Their approach differs from ours in that it explains a (fixed) external classifier and uses a predefined set of concepts, while we learn both of these intrinsically.

## 7 Discussion and future work

Interpretability and performance currently stand in apparent conflict in machine learning. Here, we make progress towards showing this to be a false dichotomy by drawing inspiration from classic notions of interpretability to inform the design of modern complex architectures, and by explicitly enforcing basic desiderata for interpretability—explicitness, faithfulness and stability—during training of our models. We demonstrate how the fusion of these ideas leads to a class of rich, complex models that are able to produce robust explanations, a key property that we show is missing from various popular interpretability frameworks. There are various possible extensions beyond the model choices discussed here, particularly in terms of interpretable basis concepts. As for applications, the natural next step would be to evaluate interpretable models in more complex domains, such as larger image datasets, speech recognition or natural language processing tasks.

**Acknowledgments**

The authors would like to thank the anonymous reviewers and Been Kim for helpful comments. The work was partially supported by an MIT-IBM grant on deep rationalization and by Graduate Fellowships from Hewlett Packard and CONACYT.

## Footnotes

[1]github.com/propublica/compas-analysis/

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
