[Supplementary Material · SENN_final_supplementary.pdf]

# A  Appendix

## A.1  Data Processing

**MNIST/CIFAR10.**  We use the original MNIST and CIFAR10 datasets with standard mean and variance normalization, using $10\%$ of the training split for validation.

**UCI.**  We use standard mean and variance scaling on all datasets and use $(80\%, 10\%, 10\%)$ train, validation and test splits.

**COMPAS**  We preprocess the data by rescaling the ordinal variable `Number_of_priors` to the range $[0, 1]$. The data contains several inconsistent examples, so we filter out examples whose label differs from a strong $(80\%)$ majority of other identical examples.

## A.2  Architectures

The architectures used for SENN in each task are summarized below, where CL/FC stand for convolutional and fully-connected layers, respectively, and $c$ denotes the number of concepts. Note that in every case we use more complex architectures for the parametrizer than the concept encoder.

|  | COMPAS/UCI | MNIST | CIFAR10 |
|---|---|---|---|
| $h(\,\cdot\,)$ | $h(x) = x$ | $\mathrm{CL}(10, 20) \rightarrow \mathrm{FC}(c)$ | $\mathrm{CL}(10, 20) \rightarrow \mathrm{FC}(c)$ |
| $\theta(\,\cdot\,)$ | $\mathrm{FC}(10, 5, 5, 1)$ | $\mathrm{CL}(10, 20) \rightarrow \mathrm{FC}(c \cdot 10)$ | $\mathrm{CL}(2^6, 2^7, 2^8, 2^9, 2^9) \rightarrow \mathrm{FC}(2^8, 2^7, c \cdot 10)$ |
| $g(\,\cdot\,)$ | sum | sum | sum |

In all cases, we train using the Adam optimizer with initial learning rate $l = 2 \times 10^{-4}$ and, whenever learning $h(\,\cdot\,)$, sparsity strength parameter $\xi = 2 \times 10^{-5}$.

## A.3  Predictive Performance of SENN

**MNIST.**  We observed that any reasonable choice of parameters in our model leads to very low test prediction error $(< 1.3\%)$. In particular, taking $\lambda = 0$ (the unregularized model) yields an unconstrained model with an architecture slightly modified from LeNet, for which we obtain a $99.11\%$ test set accuracy (slightly above typical results for a vanilla LeNet). On the other hand, for the most extreme regularization value used $(\lambda = 1)$ we obtain an accuracy of $98.7\%$. All other values interpolate between these two extremes. In particular, the actual model used in Figure 2 obtained $99.03\%$ accuracy, just slightly below the unregularized one.

**UCI.**  As with previous experiments, our models are able to achieve competitive performance on all UCI datasets for most parameter configurations.

**COMPAS.**  With default parameters, our SENN model achieves an accuracy of $82.02\%$ on the test set, compared to $78.54\%$ for a baseline logistic classification model. The relatively low performance of both methods is due to the problem of inconsistent examples mentioned above.

**CIFAR10.**  With default parameters, our SENN model achieves an accuracy of $78.56\%$ on the test set, which is on par for models of that size trained with some regularization method (our method requires no further regularization).

## A.4  Implementation and dependency details

We used the implementations of LIME and SHAP provided by the authors. Unless otherwise stated, we use default parameter configurations and $n = 100$ estimation samples for these two methods. For the rest of the interpretability frameworks, we use the publicly available `DeepExplain`[2] toolbox.

In our experiments, we compute $\hat{L}_i$ for SENN models by minimizing a Lagrangian relaxation of (5) through backpropagation. For all other methods, we rely instead on Bayesian optimization, via the `skopt`[3] toolbox, using a budget of $40$ function calls for LIME (due to higher compute time) and $200$ for all other methods.

## A.5 Qualitative results on CIFAR10

Figure 6: Explaining a CNN classifier on CIFAR10. **Top**: The attribution scores for various interpretability methods. **Bottom**: The concepts learnt by this SENN instance on CIFAR10 are characterized by conspicuous dominating colors and patterns (e.g., stripes and vertical lines in Concept 5, shown in the right-most column). All examples are correctly predicted by SENN, except the last one, in which it predicts ship instead of deer. The explanation provided by the model suggests it was fooled by a blue background which speaks against the deer class.

## A.6 Additional Results on Stability

Figure 7: The effect of gaussian perturbations on the explanations of various interpretability frameworks. For every explainer $f_{\text{expl}}$, we show the relative effect of the perturbation in the explanation: $\hat{L} = \|f_{\text{expl}}(x) - f_{\text{expl}}(\tilde{x})\| / \|x - \tilde{x}\|$.

Figure 8: **Left**: The effect of gradient-regularization on explanation stability. The unregularized version (second row) produces highly variable, sometimes contradictory, explanations for slight perturbations of the same input. Regularization ($\lambda = 2 \times 10^{-4}$) leads to more robust explanations.

## A.7 Adversarial Examples For Interpretability

We now show various examples inputs and their adversarial perturbation (accoring to (5)) on various datasets.

(A) Unregularized SENN ($\lambda = 0$)

(B) Gradient-regularized SENN ($\lambda = 5 \times 10^{-2}$)

Figure 9: Prediction explanation for two individuals differing in only on the protected variable (`African_American`) in the COMPAS dataset. The method trained with gradient regularization (column B) yields more stable explanations, consistent with each other for these two individuals.

(A) SHAP ($L = 6.78$)

(B) LIME ($L = 8.36$)

(C) SENN ($L = 0.57$)

Figure 10: Adversarial examples (i.e., the maximizer argument in the discrete version of (5)) for SHAP, LIME and SENN on COMPAS. Both are true examples are from the test fold.

Figure 11: Adversarial examples (i.e., the maximizer argument in (5)) for various interpretability methods on MNIST.

Figure 12: Adversarial examples (i.e., the maximizer argument in (5)) for SHAP, LIME and SENN on various UCI datasets.

## Footnotes

[2] `github.com/marcoancona/DeepExplain`

[3] `scikit-optimize.github.io`