[Reviews · NeurIPS 2018]

Reviewer 1



In this paper, the authors presented a framework to obtain interpretable deep networks. Inspired by the interpretability of a linear model, they 1) generalize the coefficients to be dependent on data 2) use interpretable basis concept (IBC) instead of raw pixel. The final network is similar to a two-tower model used in the recommendation system, with one tower responsible for feature extraction (regularized with reconstruction loss), the other tower responsible for supplying the coefficient (regularized with smoothness). As a result, the model approximates a linear model w.r.t. IBC locally. The authors showed their results on simple datasets. The paper is well-written and the underlying intuition is brilliant. The main drawback is the lack of a performance metric in the experimental results, e.g., the accuracy for MNIST classification. score(out of 5): quality(4) clarity(5), originality(4), and significance(4) My main concern is whether we can achieve a state-of-art performance on some of the AI task with the proposed network architecture. Since many modern networks have a linear layer as the last layer, I think the results should not be limited by the relevance tower. However, the reconstruction loss on the concept-tower might cause difficulties. It will be great if the authors can take the additional effort on obtaining a competitive result since interpretation should not just serve the purpose of interpretation: an interpretation on the best performing model can be much more meaningful. It is understandable that such tasks can be resource-demanding, so I think it should not be a deciding factor for this submission. Minor comments: Line 179: corrupted reference.

Reviewer 2



Summary: The paper proposes an alternative approach to obtaining explanations from complex ML algorithms by aiming to produce an explainable modle from the start. Recently there has been a number of works on interpretability. This work is most similar to concept-based explainability where some of the more recent ones include • Bau, David, et al. "Network Dissection: Quantifying Interpretability of Deep Visual Representations." 2017 IEEE Conference on Computer Vision and Pattern Recognition (CVPR). IEEE, 2017. • Deep Learning for Case-Based Reasoning through Prototypes: A Neural Network that Explains Its Predictions in AAAI 2018 Conditions on interpretability, as they are mentioned in this paper, have been explored earlier by f.e. Lipton, Zachary C. "The mythos of model interpretability." arXiv preprint arXiv:1606.03490 (2016)., It starts out with a linear regression model and replaces the parameters of the model with a function dependent on the input, adds an optional transformation of the input into a more low-dimensional space and a generalization of the aggregration into the output. The main novelty of this paper is the idea to start out with an intrinsically interpretable model and extending it. Review: This paper is below the acceptance threshold. They propose training an autoencoder to transform a high-dimensional input into a more interpretable feature space. To construct an appropriate distance metric for the autoencoder (i.e., one that disregards noise and compares the preservation of interpretable features), we would have to have extremely detailed knowledge of the interpretable features for the entire dataset. Otherwise there is no guarantee that the learned features are any more interpretable than the input. One of their main points is the replacements of the parameter of the linear regression model theta with a function dependent on the input. This function theta(x) is realized with a neural network, a notoriously uninterpretable algorithm. This just moves the uninterpretability but does not provide anymore insight, since the output will still depend on the (uninterpretable) NN. The authors do propose a difference boundary on theta for small differences in x. However, this restricts the interpretable part of the model to a very small input space around the current input. For interpretability we are interested in comparison to the entire data space to gain insight into why this input produces a particular output. Furthermore, they do not provide precise results or comparisons for their experiments. As an example, for MNIST it is only written that they achieve <3% error rate, which is not competitive for MNIST. For UCI they only mention ‘competitive performance’, when the results could easily have been included in the supplementary. As such, I would consider the experiments insufficient. Additionally, harder tasks are not included in the experiments. Some ideas in the paper are promising and I recommend the authors submit this paper to a workshop. The paper is written clearly and concisely. The idea of generalizing an interpretable model into a more complex one is interesting and to my knowledge novel. In the same vein, the proposed conditions for interpretability have not been published in this form before. In my opinion, more experiments with more complex datasets would greatly benefit this paper as it is unclear to me whether the approach generalizes to more complicated tasks. Minor comments: L. 179: wrong ref approach ?? L. 213 wrong ref

Reviewer 3



This paper introduces a self-explaining model which progressively generalizes linear classifiers to complex yet architecturally explicit models. Two parallel networks are utilized to generate concepts and relevances, respectively, then a linear aggregator is applied to do the classification. Three desiderata for explanations are proposed, including explicitness, faithfulness, and stability. Experiments on three datasets (including MNIST, UCI datasets, and Propublica’s COMPAS Recidivism Risk Score dataset) are conducted to demonstrate the explicitness/intelligibility qualitatively, the faithfulness and stability quantitatively. I have several comments to this work: 1. The model proposed in this paper seems novel and interesting. Different from the posteriori explanations for previously trained models, the authors introduce self-explaining model trying to achieve explicitness, faithfulness, and stability for predictions locally. However, there is a key point for self-explaining model: can the self-explaining model achieve comparable prediction or classification accuracy to the black-box deep model, and at the same time be more interpretable? If not, what are the advantages of the self-explaining model compared with the posteriori explanations? Could the authors explain this and show more experimental results to demonstrate the prediction accuracy of the self-explaining model? 2. The authors indicate that pixels are rarely the basic units used in human image understanding, instead, people would rely on higher order features. However, in the experiments, the authors use only MNIST dataset which is simple for vision task. Other datasets (e.g., CIFAR-10) are expected in the explicitness/intelligibility to demonstrate whether the proposed SENN can generate these higher order concepts.